# Randomised controlled pilot feasibility trial of an early intervention programme for young infants with neurodevelopmental impairment in Uganda: a study protocol

Margaret Nampijja,[1] Emily Webb,[2] Carol Nanyunja,[1] Samantha Sadoo,[3] Ruth Nalugya,[1] James Nyonyintono,[4,5] Anita Muhumuza,[6] Moses Ssekidde,[5] Kenneth Katumba [ORCID] ,[1] Brooke Magnusson,[4] Daniel Kabugo,[4] Frances M Cowan,[7] Miriam Martinez-Biarge,[7] Maria Zuurmond,[8] Cathy Morgan,[9,10] Deborah Lester,[4,11] Janet Seeley,[1,12] Cally J Tann[1,3,13]

For numbered affiliations see end of article.

**Correspondence to**
Dr Cally J Tann;
Cally.Tann@lshtm.ac.uk

## ABSTRACT

**Introduction** Early intervention programmes (EIPs) for infants with neurodevelopmental impairment have been poorly studied especially in low-income settings. We aim to evaluate the feasibility and acceptability of a group participatory EIP, the 'ABAaNA EIP', for young children with neurodevelopmental impairment in Uganda.

**Methods and analysis** We will conduct a pilot feasibility, single-blinded, randomised controlled trial comparing the EIP with standard care across two study sites (one urban, one rural) in central Uganda. Eligible infants (n=126, age 6–11 completed months) with neurodevelopmental impairment (defined as a developmental quotient <70 on Griffiths Scales of Mental Development, and, or Hammersmith Infant Neurological Examination score <60) will be recruited and randomised to the intervention or standard care arm. Intervention arm families will receive the 10-modular, peer-facilitated, participatory, community-based programme over 6 months. Recruited families will be followed up at 6 and 12 months after recruitment, and assessors will be blinded to the trial allocation. The primary hypothesis is that the ABAaNA EIP is feasible and acceptable when compared with standard care. Primary outcomes of interest are feasibility (number recruited and randomised at baseline) and acceptability (protocol violation of arm allocation and number of sessions attended) and family and child quality of life. Guided by the study aim, the qualitative data analysis will use a data-led thematic framework approach. The findings will inform scalability and sustainability of the programme.

**Ethics and dissemination** The trial protocol has been approved by the relevant Ugandan and UK ethics committees. Recruited families will give written informed consent and we will follow international codes for ethics and good clinical practice. Dissemination will be through peer-reviewed publications, conference presentations and public engagement.

**Trial registration number** ISRCTN44380971; protocol version 3.0, 19th February 2018.

## Strengths and limitations of this study

► This pilot feasibility trial is among the first to examine feasibility and acceptability of an early intervention programme for young children with neurodevelopmental impairment in a low-resource sub-Saharan African setting.

► The mixed-method evaluation of this complex community-level intervention will provide important information on implementation of an early intervention programme for child disability at scale.

► While the small sample size and individually randomised trial design will limit our understanding of programme impact, quantitative and qualitative data will inform design and execution of a larger future trial to examine effects on important child and family outcomes.

## INTRODUCTION

Globally each year, an estimated 30 million neonates experience complications around the time of birth which can have a life-long impact on health and development.[1] The United Nations Global Strategy for Women's, Children's and Adolescents' Health (2016–2030) advocates the need for all children to 'survive' and to 'thrive'.[1] While in recent decades substantial progress has been made in reducing child mortality in low-income and middle-income countries (LMICs), the global burden of developmental disabilities remains unchanged.[2] Child neurodevelopmental impairment (NDI) significantly impacts families in any context, but particularly in low-resource settings, where availability and access to support services are limited, financial barriers greater and social stigma more overt.[3]

A wide spectrum of impairment is seen after newborn illnesses, including cerebral palsy, ineffective feeding, learning, visual and hearing difficulties and epilepsy.[4] A growing evidence base, largely from high-income countries (HICs), suggests that early intervention programmes (EIPs) commencing in the first months after birth have the potential to limit and even prevent developmental and cognitive impairments following early brain injury. These programmes target the neuroplasticity of the immature developing brain, either directly or indirectly, through family capacity building and enrichment of the care-giving environment.[5]

In HICs, it has been shown that early environmental enrichment can enhance motor function in children<2 years.[4 6] In LMICs, several trials have shown positive effects of EIPs in at-risk infants,[7–10] although these studies have not focused on infants specifically with NDI. Few studies have examined the feasibility and acceptability on affected children and their caregivers, and how they might be integrated into existing community health programmes.[11] Scalability and sustainability of an intervention programme are also dependent on its cost effectiveness. This is particularly true in LMICs where resources are scarce and existing care structures for children much less well established.

## Aims

The study aims to evaluate whether a facilitated, community-based, participatory EIP is feasible and acceptable. We will conduct a pilot feasibility single-blind, randomised controlled trial (RCT) with two parallel groups. The outcomes of interest are feasibility of randomisation and recruitment, acceptability among caregivers and healthcare workers and early evidence of family impact quality of life (QoL), 6 months after recruitment and again 6 months later. The incremental and protective cost effectiveness of the EIP and the economic impact of child developmental disability to families and services in Uganda will be examined by Katumba *et al* in a separate protocol.

## Objectives and hypotheses

The primary objectives of the study are to

1. Describe the feasibility and acceptability of the EIP for children with NDI and their families.
   *Hypothesis: It will be feasible to conduct an RCT of the EIP versus standard care (SC) in rural and urban contexts and acceptable to families and the community.*
2. Obtain preliminary data on whether the EIP improves family QoL when compared with SC.
   *Hypothesis: Families receiving the community-based EIP will demonstrate improved QoL scores on the Paediatric Quality of Life Family Impact module compared with SC 12 months after recruitment.*
3. Identify the main barriers and facilitating factors for scaling up the EIP.
   *Hypothesis: The EIP is scalable in this low-resource Ugandan setting.*

4. Determine the incremental and protective cost effectiveness of the ABAaNA EIP.
   *Hypothesis: The ABAaNA EIP is a cost-effective intervention to improve family QoL for children with NDI.*

## METHODS

We will conduct a pilot feasibility, single-blind, RCT with two parallel groups: one receiving the EIP and the other SC.

## Study setting

The study is based at two Ugandan sites: one urban (Mulago Hospital, Kampala) and one rural (Kiwoko Hospital, Nakaseke). Neither site has existing family support services for children with NDI.

Mulago National Referral Hospital is the largest in Kampala, Uganda's capital city, taking high-risk pregnancies from across surrounding areas. Children's services include acute admissions, an inpatient malnutrition unit and outpatients, with a weekly paediatric neurology clinic providing investigation and management of neurological conditions including seizures, and a clinic-based physiotherapy and occupational therapy service for children with cerebral palsy and other NDIs.

Kiwoko Hospital in Nakaseke District, central Uganda, serves a catchment area of 800 000 people and provides comprehensive medical services, including neonatal inpatient care for >1200 infants per year. The trial implementation partner, Adara Development, has worked in partnership with Kiwoko Hospital since 1998, and the government to improve neonatal health in Nakaseke district. Together they provide HIV services, maternal health services and community-based healthcare to 44 villages surrounding Kiwoko Hospital.

## Participants and recruitment

Participants will be young children with NDI and their caregivers. A Standard Protocol Items: Recommendations for Interventional Trials diagram showing the planned flow of participants is presented in figure 1.

### Screening for eligibility

Infants at high-risk of NDI will be identified from (1) neonatal admission registers and neonatal follow-up services, (2) local paediatric outpatient services and (3) attendance for early child health services following community sensitisation. Sensitisation will include public health announcements on local radio raising awareness of the research and appropriate child development more generally. Caregivers of high-risk infants (survivors of neonatal encephalopathy, prematurity, neonatal septicaemias/meningitis and severe jaundice) will be contacted by phone and invited to attend an appointment when the child is 6–11 completed months old. After informed written consent, they will be screened for NDI by trained study staff using the Malawi Developmental Assessment Tool (MDAT).[12] If two or more items in any MDAT domain are not achieved, the child will be referred for comprehensive

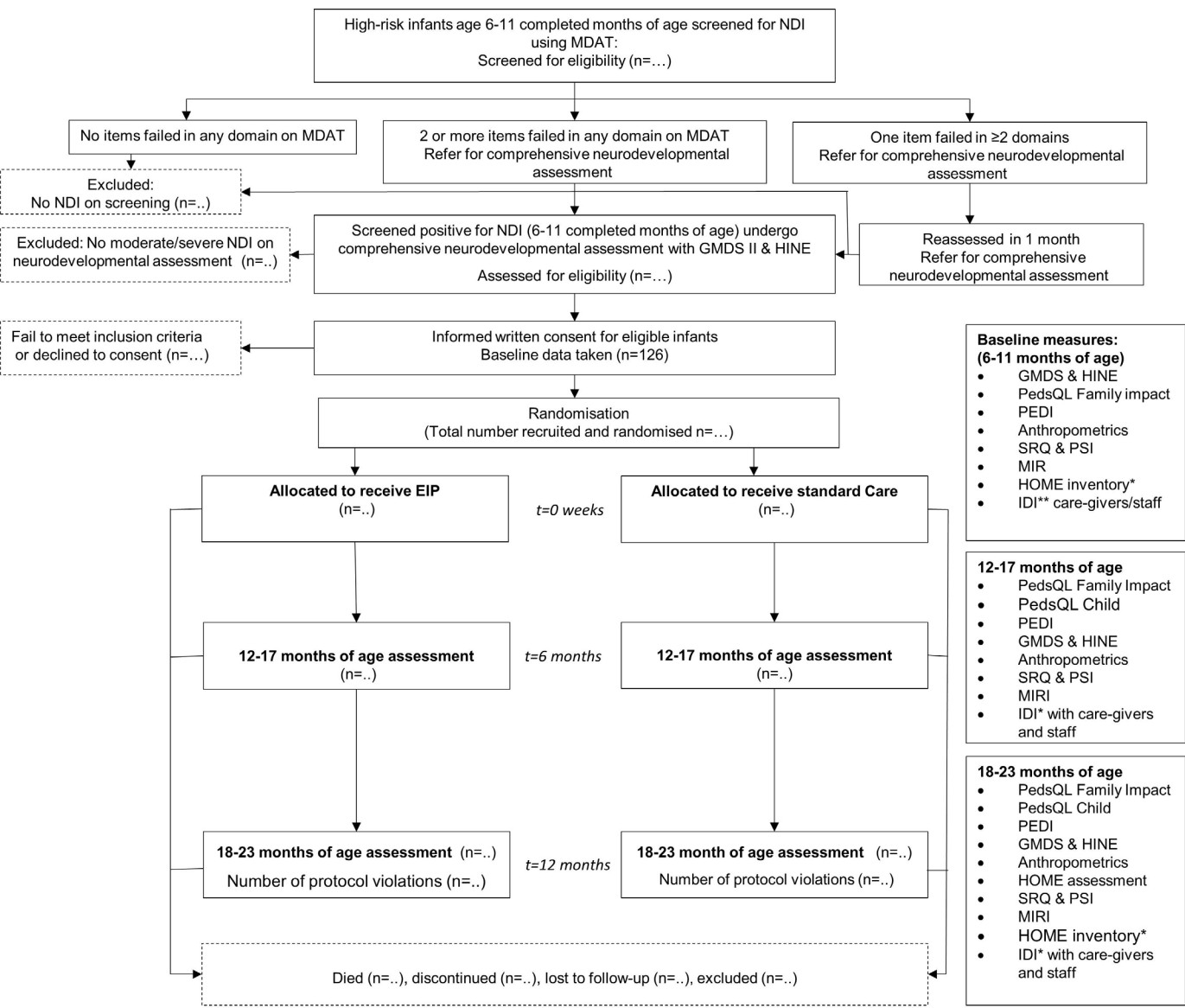

**Figure 1** Flow of participants. *In-depth interviews (IDI) with caregivers on impact of disability, confidence level of the parents, level of participation in family and community life and experience of stigma/discrimination. EIP, early intervention programme; GMDS, Griffiths Mental Developmental Scales; HINE, Hammersmith Infant Neurological Examination; HOME, Home Observation for the Measurement of the Environment; MDAT, Malawi Developmental Assessment Tool; MIRI, Maternal Infant Responsiveness Inventory; NDI, neurodevelopmental impairment; PedQL, Pediatric Quality of Life tool; PEDI, Pediatric Evaluation Disability Inventory; PSI, Parent Stress Index; SRQ, Self-Referral Questionnaire.

neurodevelopmental assessment. If the child fails one item in two or more domains, they will be invited back for an assessment in 1 month. If the child's MDAT scores are age appropriate across all domains, advice will be given on play and stimulation, communication, nutrition and immunisations and the child discharged.

Caregivers of infants screening positive on MDAT will be invited to an appointment for written informed consent, and if provided, comprehensive neurodevelopmental assessment by study staff using the Griffiths Mental Developmental Scales (GMDS)[13] and the Hammersmith Infant Neurological Examination (HINE).[14] An overall developmental quotient will be derived, from the GMDS subscales assessing locomotor, personal–social, hearing/

language, eye-hand coordination and performance skills.[13] Neuromotor impairment will be further assessed according to the HINE, a standardised paediatric neurological examination and classified by type. We have used both these tests extensively in previous studies in Uganda and found them easy to administer in this setting and at this age.[15] The assessments will be conducted in the local language using the standard manual material to ensure internal consistency in the assessments technique. Inclusion and exclusion criteria are outlined in box 1.

### Baseline characteristics
Infant and caregiver demographic information will be recorded at baseline, including date of birth, age, sex,

---

**Box 1  Eligibility for inclusion in the randomised controlled trial**

**Inclusion criteria**
► Infant aged 6–11 completed months
► Moderate–severe neurodevelopmental impairment (NDI) defined as a Griffiths Mental Developmental Scales (GMDS) Developmental Quotient <70 and/or Hammersmith Infant Neurological Examination (HINE) score <60 (Romeo, 2013)
► Informed written consent by caregiver

**Exclusion criteria**
► Infants aged 12 months of age or older
► Infants screening positive for NDI (using Malawi Developmental Assessment Tool) but not meeting the criteria for moderate–severe NDI on GMDC and HINE assessment
► Conditions requiring prolonged inpatient treatment
► Parents unwilling or unable to attend the full programme
► Main residence outside Nakaseke or Luwero district, and >20 km from Mulago Hospital
► Accompanying caregiver not speaking or understanding Luganda or English

---

**Box 2  Developing the ABAaNA early intervention programme (EIP)**

► In low-income and middle-income countries, services for affected children are often lacking and parental levels of knowledge and understanding about cerebral palsy are often low. To fill this gap, a parent training programme called 'Getting to Know Cerebral Palsy' was developed and launched in partnership between the London School of Hygiene & Tropical Medicine and Christian Blind Mission an international disability and development organisation. The programme aims to increase parental knowledge and skills and promotes a participatory learning approach with an emphasis on the empowerment of caregivers across a broad spectrum of impairment for children aged 2–12 years.[23 24]
► Since 2011, the ABAaNA studies ('Abaana', meaning 'children' in the local language Luganda) have been examining risk factors for, and outcomes from, neonatal encephalopathy (NE) in Uganda.[15] Studies examining early neurodevelopmental outcomes after NE revealed a high prevalence of neurodevelopmental impairment (NDI) with 25% of those affected also having malnutrition from related feeding difficulties.[15] Qualitative work highlighted the stigma and broad-ranging social, emotional and financial impacts on affected families.[3]
► In response, the ABAaNA EIP was developed around the principles of 'Getting to Know Cerebral Palsy' (http://www.ubuntu-hub.org), and has been adapted for younger children aged 0–2 years following an iterative process following Medical Research Council recommendations on development and evaluation of complex interventions[25]; it was supported by a diverse Expert Advisory Group including local parents with children with NDI, Disabled Persons Organisations and experts in early intervention and child development. Core themes running through the programme are summarised in figure 2. The newly developed programme was piloted among 28 families at Mulago Hospital in Kampala in 2015–2016 and showed a 25% improvement in family quality of life scores (Paediatric Quality of Life tool, Family *Impact* module 2.0) post intervention (verbal communication).

---

birth order, parity, antepartum, intrapartum and postpartum history, family and medical history, developmental history, mother's education and occupation, family details including family size, and ages, household incomes, household SES and residence. All outcome measures will also be measured at baseline enabling preintervention and postintervention comparisons.

### Randomisation
Infants and their caregivers will be randomised in a 1:1 ratio to either the EIP or SC arm. Randomisation will be stratified by recruitment centre. Randomisation lists indicating a randomisation number and trial arm allocation will be prepared by the trial statistician using a random number generator in Stata (V.15) prior to the commencement of the study, and stored on a secure, password-protected computer at the Medical Research Council/Uganda Virus Research Institute (MRC/UVRI) and London School of Hygiene & Tropical Medicine (LSHTM) Uganda Research Unit by a statistician otherwise not involved in the study. When a participant is eligible for recruitment and consent obtained, study staff will contact the MRC/UVRI statistician who will inform the study staff of the study number and trial arm to which the participant is to be allocated. The personnel in charge of the randomisation will not be involved in other study procedures, including assessment of outcomes.

### Intervention arm
The EIP is a community-based, peer-led group programme with caregivers at a community level, using a participatory approach driven by adult learning theory.[16] The programme manual is freely available to download (https://www.ubuntu-hub.org). Development of the programme is described in box 2.

Participating families are encouraged to share experiences through discussion and reflection, prioritise problems and identify solutions together. Facilitators of the group sessions are 'expert parents', themselves parents of children with NDI, who have undergone 5 days of core training followed by regular supervision, face-to-face mentoring meetings and telephone discussions with existing in-country master facilitators (trained therapists in Uganda). Each EIP group involves 6–10 families; groups are selected depending on locality for ease of attendance. The training is divided into 10 modules covering understanding disability, positioning and carrying, feeding, mobilising, communication, play, everyday activities and experiences in the local community (figure 2, table 1). Individual module sessions are delivered every 1–2 weeks and last 2–3 hours including time for facilitated discussion; the entire programme is designed to be delivered over 6 months including at least one home visit conducted by the expert parent facilitator.

### Fidelity and adherence to the intervention
EIP facilitators will receive a 5-day training programme delivered by two master facilitators, which includes

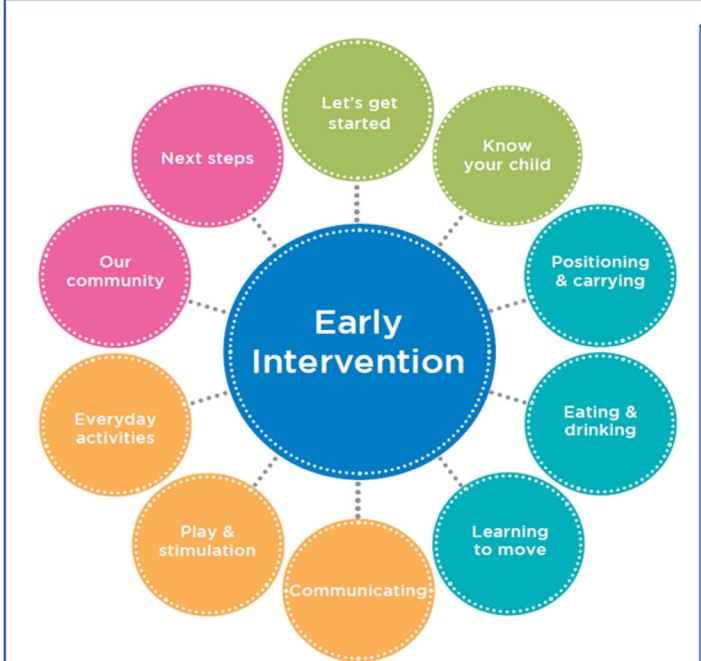

**Figure 2** Core themes and content of the ABAaNA early intervention programme.

facilitation skills, knowledge transfer on the core contents of the EIP manual and translation of knowledge to practice through simulated sessions with families and children with NDI. All trial intervention groups will be co-facilitated by a master facilitator providing supportive supervision to new facilitators. After each modular group meeting, a short-facilitated feedback session will be conducted, and the content of the module delivered will be recorded. Attendance of individual caregivers and children at the group sessions will be recorded. Facilitators will emphasise to caregivers the importance of attending all sessions, with phone calls prior to each session to promote adherence. If missed, a catch-up session may be offered before the next module.

### Standard care arm

SC refers to care that is currently available in established local services. In both sites, this includes referral to physiotherapy, seizure management and nutritional support. Information on access to local medical, therapy and family services will be collected. Families in the SC arm will be offered delayed entry into the EIP after completing the 18-month assessment. Contamination of the SC arm by exposure of SC families to intervention will be monitored and reported.

### Outcomes

Participants in both arms will be assessed by study staff masked to trial allocation at two time points; at age 12–17 months (which corresponds to completion of the EIP in the intervention arm, 6 months after recruitment) and again at age 18–23 months (12 months after recruitment). (figure 1). Caregivers will be phoned a week before the follow-up assessments to arrange a time for

interview. Assessments will be primarily conducted in the study-site clinics. Where caregivers cannot be contacted by phone or are unable to attend the clinic, a community visit will be arranged, and assessments completed at home. Outcome assessments will be conducted by Mulago assessors for children recruited at Kiwoko, and vice versa to ensure assessors are blind to allocation arm. Two assessors will independently assess a small proportion of the children and inter-rater reliability will be calculated.

### Primary outcome measures

The primary outcomes of the study will be:

1. *Feasibility of participant recruitment and randomisation* as assessed by the total number recruited and randomised to each arm. Qualitative tools will also be used to capture information on feasibility.
2. *Acceptability of the EIP among caregivers and healthcare workers* as assessed by the protocol violation rate (eg, participants in the intervention arm being treated as if they were in the control arm or vice versa) at programme completion, and by the number of programme sessions attended between baseline and programme completion. Qualitative tools will also be used to capture information on acceptability.
3. *Preliminary evidence of impact on family QoL* as assessed using the scored Paediatric Quality of Life (PedsQL) Family Impact module.[17] The PedsQL comprises 36 items scored on a 0–4 Likert scale and linearly transformed to a 0–100 scale, with higher scores indicating a better QoL. It will be translated into the local language Luganda and administered as a standardised structured interview by trained study staff.

| Table 1 | Description of the programme modules |
|---|---|
| **Module** | **Content** |
| 1.Let's get started | Content and ground rules of the programme<br>Understanding cerebral palsy, additional resources for information<br>Personal stories |
| 2.Know your child | Developmental milestones for young children<br>Determining each child's progress |
| 3.Positioning and carrying | The importance of optimal positioning<br>Practical skills regarding optimal positioning |
| 4.Eating and drinking | Feeding challenges for children with neurodevelopmental impairment<br>Practical skills for addressing feeding challenges |
| 5.Learning to move | Understanding different types of movement<br>Practical skills for assisting learning to move |
| 6.Communicating | The importance of communication<br>Practical advice to encourage their child to communicate |
| 7.Play and early stimulation | The importance of early stimulation and play for children to develop<br>Challenges of inclusion in play with the family and community<br>Creation of simple toys<br>How parents/ caregivers can encourage their child to play |
| 8.Everyday activities | Using everyday activities to promote child development<br>Management of seizures<br>Review of previous sessions |
| 9.Our community | Community resources available<br>Discussion around barriers to inclusion, addressing stigma and discrimination<br>Understanding disability rights<br>Thoughts and feelings of the caregiver<br>Members of community invited to attend this session |
| 10.Next steps | Planning to facilitate their own group<br>Reflection on learning points<br>Endpoint data collection |

### Other outcomes of interest

1. *Child motor functioning* as assessed by the mobility score of the Paediatric Evaluation Disability Inventory (PEDI).[18] The PEDI is a standardised test designed to identify and describe functional impairment and monitor progress. Normative scaled scores are obtained for children≥6 months to provide age-related expectations of ability.
2. *Child cognitive function* as assessed by the GMDS.[13]

3. *Child growth, health and well-being* assessed using weight, height and head circumference measured according to standardised protocol. Occipitofrontal head circumference (paper tape measure), weight (SECA336 electronic scales, Hamburg, Germany) and height will be taken by study staff using standardised procedures. Haemoglobin will be determined on a finger prick sample using HemoCue Hb 201 (HemoCue AB, Angelholm, Sweden). A structured maternal interview in Luganda will report on caregivers concerns regarding health, growth and development of their child and episodes of illness including seizures and other neurological problems, feeding difficulties, chest infections and treatment for malnutrition.
4. *Caregiver psychological distress* assessed using the Self-Referral Questionnaire (SRQ) and the Parenting Stress Index (PSI).[19] The SRQ consists of 20 items each scored 0 (symptom absent) or 1 (symptom present) giving a total out of 20. The PSI is a 120-item inventory measuring the magnitude of caregiver stress attributable to parent–child relationship (Total Stress Scale), and situational/demographics factors outside the parent–child relationship (Life Stress Scale). These tools will be translated into Luganda.
5. *Caregiver–child attachment* assessed using the Maternal Infant Responsiveness Instrument; a 22-item scale designed to measure the parent's feelings about their infant and an appraisal of the infant's responses.[20]
6. *Quality of the home environment* assessed using the Infant Toddler-Home Observation for the Measurement of the Environment. This comprises 45 items, based on observation and/or interview, assessing the physical environment of the home and the child's interaction within it.[21]
7. *Cost of illness and protective effectiveness* will be assessed (separate protocol, Katumba *et al*).

### Qualitative methods

In-depth interviews (IDIs) will be conducted with five randomly selected caregivers from each arm at each site. Focus group discussions (FGDs) will be conducted with caregivers, at baseline, 6 months post recruitment and again 6 months later in both the intervention and SC arms. Among intervention arm families, qualitative techniques will be used to capture information on the feasibility, acceptability and impact of the EIP intervention using qualitative tools including FGDs, IDIs and observation.

We will describe the experiences of children and caregivers relating to the intervention received including the impact of the disability, parental confidence level, inclusion in community life and experience of stigma and discrimination. We will examine changes in these domains over the follow-up period and explore attributions of change. In addition, we will perform social mapping of parent networks and group discussions with staff on their perspectives and experiences of using the EIP. The themes guiding our analysis will be drawn from objectives

of the trial and from the data, should additional areas of interest emerge during interviews and discussions.

The interviews will be conducted by social scientists who have experience in qualitative research.

### Data management and access

Data collected in the clinic or at field visits will be entered on standard clinical record forms (CRFs). Clinical data will be recorded under a unique study ID number. Completed CRFs will be checked by and double entered into a trial-specific MS Access database. Data from both IDIs and FGDs will be collected in the form of audio-tapes, transcripts and field notes. All data entry and data management will be overseen by a statistician/data manager at the MRC/UVRI Unit. Data will be maintained on the host institution server and backed up following standardised operating procedures. Paper CRFs will be stored in lockable filing cabinets at the sites. Access to these data during the trial will be restricted to essential personnel (the principal investigators, site co-investigators, medical research officers and data clerks).

### Confidentiality

All research team members will receive training in confidentiality. Data will be stored without personal identifiers, except where names must be included to ensure identification of the correct participants for procedures. All data will be stored on password-protected computers, accessible only to research team members.

### Sample size

The trial will recruit 126 children and their caregivers, 63 per arm. Allowing for a 20% dropout rate, this sample size will give 90% power to detect a minimal relative difference of 20% on PedQL Family Impact score between the intervention and control arms, at 5% significance level, assuming a mean PedQL score of 65 in the SC arm and SD of 20 in both arms. Assumptions are based on data from the pilot study showing a mean caregiver PedQL score for families before the intervention of 64.9 (SD 19.6) and mean score of 78.9 for families after receiving the intervention (SD 17.5).

### Statistical analysis

The first primary outcome, feasibility of participant recruitment and randomisation, will be assessed by the total number recruited and randomised to each arm. Recruitment and randomisation feasibility will be demonstrated if the target sample size of 126 is achieved. Data on participants screened, eligible and randomised will be displayed in a Consolidated Standards of Reporting Trials flow chart. Descriptive statistics (frequencies, means, medians, SD and IQRs) will be used to describe the sample at baseline, by trial arm.

The second primary outcome, acceptability, will be assessed quantitatively by (1) calculating the protocol violation rate and (2) summarising the number of programme sessions attended between baseline and programme completion for those in the intervention arm.

Protocol violation rate will be calculated as the number of participants for whom one or more protocol violations occur divided by the total number of participants, and will be presented both overall, and by trial arm. For participants in the EIP trial arm, the overall number of modules attended by each participant will be tabulated. Acceptability on the basis of number of programme sessions will be defined as attendance of at least six modules.

For the third primary outcome and secondary outcomes, analyses will compare outcomes between intervention and control arms at the end of the programme, when the participants will be aged 12–17 months, and again 6 months later. Analysis will be on an intention-to-treat basis and missing data will not be imputed. Data for each outcome measure will be summarised by trial arm, using proportions for binary outcomes and means or medians for quantitative outcomes, depending on normality of the distribution. Differences in means/proportions between trial arms together with 95% CIs will be calculated. We do not plan any formal statistical tests due to the preliminary nature of the trial; instead CIs will provide a possible range of effect sizes. Regression models (linear regression for continuous outcomes, logistic regression for binary outcomes) will be used to adjust comparisons for baseline measures of the outcomes, which were collected at enrolment into the trial, in order to improve precision of effectiveness estimates. For skewed continuous outcomes, data will be normalised before analysis using suitable transformations or quantile regression will be considered. No subgroup analyses are planned.

Qualitative data will be analysed using a thematic framework approach. Themes will be based on the study objectives and those emerging from the data. Social scientists (two people) will agree the coding frame and undertake analysis collaboratively to ensure agreement on the coding approach. Thematic summaries will be developed and shared with the wider team for discussion.

### Trial management, data monitoring and reporting of adverse events

The Trial Steering Committee (TSC)[22] will oversee progress of the study towards its objectives, review relevant information from other sources (eg, other related trials) and receive reports from the Data and Safety Monitoring Board (DSMB). All adverse events, whether related to the intervention or not, will be noted and reported. A Data Monitoring and Safety Committee has been established independent of the investigators and the TSC but reporting to the TSC and the sponsor. The DSMB includes an expert on global child heath, a senior statistician and a senior academic working in newborn and early child health research in Uganda, independent of the investigators. The DSMB will have access to all data on request. Resulting from the initial meeting of the DSMB on 28 June 2017, no formal stopping rules will be applied.

Children with NDI and particularly those with seizure disorders and difficulties with swallowing are at increased mortality risk. All adverse events, whether related to the

intervention or not, will be investigated and reported according to the UVRI Research Ethics Committee (REC) in accordance with good clinical practice requirements. All deaths, hospitalisations and other serious adverse effects will be reported to the relevant ethics committee irrespective of whether the death or event is related to disease progression or not. Trial data monitoring will be conducted by an internal independent monitor at initiation, 6 months into data collection, again after 1 year and end of data collection.

## PARTICIPANT AND PUBLIC INVOLVEMENT

The intervention, study design and conduct were developed directly from the engagement of caregivers and programme facilitators ('expert parents') with a parent representative on the TSC. The priorities and experiences of caregivers identified during facilitated group discussions at a key-stakeholders meeting (June 2017) contributed to the development of our research question and outcome measures. Plans to communicate findings to participants and the wider community will involve caregivers, through formal discussions with the TSC.

## ETHICS AND DISSEMINATION
### Ethics

The protocol has been approved by the Research and Ethics committee of the UVRI, Mulago Hospital and Kiwoko Hospitals, the Uganda National Council for Science and Technology, the Uganda President's Office, and the ethics committee of the LSHTM. Information sheets will be available in English and Luganda, the main local language. Parents will be provided with an oral and written explanation of the study by Ugandan study staff to ensure that information is accessible to those with lower levels of literacy. Witnessed consent using a thumb print will be available to parents/guardians who are non-literate. Reimbursement for the cost of transport will be provided to caregivers on attendance at the screening and recruitment visits.

All recruited children will receive SC at the study sites. This will include referral to local services for seizure management and physiotherapy where available. To date, the benefits of the proposed EIP have not been proven and may have a negative effect if children are incorrectly classified as having NDI and placed in the programme. Children and their caregivers in the control arm will receive delayed entry into the programme for older children ('Getting to Know Cerebral Palsy') at 18–23 months at the time of their final study assessments.

### Dissemination

Our programme has strong links with partnership organisations working in Maternal and Child Health programming including Adara Development, Kiwoko Hospital, Nakaseke District Health Office and other collaborating institutions. Research findings will be disseminated to the Ministry of Health, to inform local and national health policies. Regional-level stakeholders, including the Nakaseke District Health Office and heads of regional health and social services, will be engaged to support staff recruitment, contributing to the sustainability of the innovation at local and district levels. Meetings for key stakeholders, including local Non-Governmental Organisations working in child disability will be held twice during the project period to promote buy-in, facilitate fast-cycle learning, disseminate study findings and ultimately promote sustainability of the programme. Global learning will be facilitated through our existing online community of practice spanning 70 countries and >300 members. Communications support staff at MRC/UVRI, LSHTM and Adara Development will facilitate dissemination of information through appropriate media outlets, the web and social media.

Study findings will be published through Open Access peer-reviewed journals, presentations at local, national and international conferences and to the local community through community meetings. Written reports will be submitted to UVRI REC and reported to the trial registry. Data will be made available on request.

**Author affiliations**
[1]Social Aspects of Health Programme, MRC/UVRI & LSHTM Uganda Research Unit, Entebbe, Uganda
[2]MRC Tropical Epidemiology Group, London School of Hygiene and Tropical Medicine, London, United Kingdom
[3]Department of Infectious Disease Epidemiology, London School of Hygiene and Tropical Medicine, London, United Kingdom
[4]Adara Development, Washington, United Kingdom
[5]Kiwoko Hospital, Nakaseke, Uganda
[6]Neonatal Medicine, Mulago National Referral Hospital, Kampala, Uganda
[7]Department of Paediatrics, Imperial College London, London, United Kingdom
[8]Faculty of Infectious and Tropical Diseases, London School of Hygiene and Tropical Medicine, London, United Kingdom
[9]Cerebral Palsy Alliance Research Institute, Sydney, New South Wales, Australia
[10]Paediatrics and Child Health, University of Sydney, Sydney, New South Wales, Australia
[11]Seattle Children's Hospital, Seattle, Washington, USA
[12]Faculty of Public Health and Policy, London School of Hygiene and Tropical Medicine, London, United Kingdom
[13]Neonatal Medicine, University College London Hospitals NHS Trust, London, United Kingdom

**Acknowledgements** The authors would like to thank the members of the ABAaNA Early Intervention Programme Expert Advisory Committee who supported the development of the programme and the implementation partners Kiwoko Hospital, Adara Development and Mulago University Hospital. They would like to acknowledge the contribution of Ellipsis for formatting the programme manual and thank Joanna Lawn for her contribution to manual illustrations. The authors would like to thank the funders, Grand Challenges Canada, Saving Brains and the Research Foundation of Cerebral Palsy Alliance Australia.

**Collaborators** Giulia Greco, Heidi McNamara, Carol Nanyunja, Christine Otai, Margaret Musoke, Margaret Sewegaba, Julius Ssekyewa

**Contributors** The study was conceived and designed by CJT with substantial contribution from MN, DL, JN, EW, CM, JS, KK and FMC. Research methodology was developed by MN, CN, MZ, JN, BM, DK, SS, RN, MS, FMC, MM-B, AM and CJT. The first version of the paper was written by CJT, SS and MN. All authors contributed to the final version of the manuscript.

**Funding** This work is supported by Saving Brains, Grand Challenges Canada grant number 1707-08687. The economic evaluation is supported by a Project Grant, awarded by the Research Foundation, Cerebral Palsy Alliance (PG02917). The trial

sponsor is the MRC/UVRI and LSHTM Uganda Research Unit, Entebbe; Contact name: Pontiano Kaleebu (Director); Address: MRC/UVRI & LSHTM Uganda Research Unit, Plot 51-59 Nakiwogo Road, P.O. Box 49 Entebbe, UGANDA Tel: +256 (0) 417 704000; mrc@mrcuganda.org.

**Disclaimer** The funder had no role in the research design and will not have any role in the execution, analyses, interpretation of the data or decision to submit results.

**Competing interests** None declared.

**Patient consent for publication** Not required.

**Ethics approval** The protocol has been approved by the Research and Ethics committee of the UVRI, Mulago Hospital and Kiwoko Hospitals, the Uganda National Council for Science and Technology, the Uganda President's Office, and the ethics committee of the LSHTM.

**Provenance and peer review** Not commissioned; externally peer reviewed.

**ORCID iD**
Kenneth Katumba http://orcid.org/0000-0002-6726-740X

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
