## [Reviewer comments · BMJ Open]

ARTICLE DETAILS

TITLE (PROVISIONAL)	A Randomised Controlled Pilot Feasibility Trial of an Early Intervention Programme for Young Infants with Neurodevelopmental Impairment in Uganda: Study Protocol
AUTHORS	Nampijja, Margaret; Webb, Emily; Nanyunja, Carol; Sadoo, Samantha; Nalugya, Ruth; Nyonyintono, James; Muhumuza, Anita; Ssekidde, Moses; Katumba, Kenneth; Magnusson, Brooke; Kabugo, Daniel; Cowan, Frances; Martinez-Biarge, Miriam; Zuurmond, Maria; Morgan, Cathy; Lester, Deborah; Seeley, Janet; Tann, Cally

VERSION 1 – REVIEW

REVIEWER	Maiken Pontoppidan The Danish Center for Social Science Research Denmark
REVIEW RETURNED	25-Jul-2019

GENERAL COMMENTS	This is a trial protocol for a study examining feasibility, acceptability and the effects of the early intervention programme ABAaNa for parents of infants with a neurodisability in Uganda. The protocol is well written but tends to be a bit long with some repetitions. Major issues: The protocol does not appear to be rewritten to a journal paper. E.g. there is no abstract and there is a content list. I do not find any reference to a registration in a trial register. Has the trial been registered prior to recruitment? There is no actual status of the trial. According to the timeline recruitment ended in 2018 - if this is true the trial is not currently recruiting. SPIRIT checklist: please make sure that the checklist refers to the right pages. E.g. it says that trial identifier and registry name is on page 2, Sample size calculation is on p 25 (12-13 in the checklist); analysis plan is on p 26 (13 in the checklist) Please add line numbers There are some repetitions - e.g. section 5.1.3 is a longer version of 3.4; section 5.2.4 is a longer version of an earlier; section 5.3 is a longer veriosn of 3.6.3 - It would be easier if it is merged so all the information is at one place. P 22: 5.2.6, 5.2.7, 5.2.8 who does the assessment? interview or self-report? Please add information on how training of the persons doing the intervention will be carried out. Analysis: do you plan to di any imputation of missing data? Minor issues:
--

	Please add the references to paragraphs 2, 3 and 4 on p 7, 4 on p 9, 3 on p 10, P 10 paragraph 3: Are there no never or ongoing studies? Maybe some of the studies by Lynne Murray and Mark Tomlinson could be relevant? All tables: please spell out abbreviations in a note. E.g. Table 1 DQ and HINE. All tables: please indicate if high or low score is good Table 2: what is N? can you add effect sizes? P 15 paragraph 5: who does the screening? p 19 3.6.1 QOL - is it parent report? P 19 3.6.2 How will outcomes be assessed? Please add expected effect size to sample size calculation p 25 P 26 qualitative data - please add reference to thematic framework approach and add i bit more information on which areas or themes you expect to analyse
--	---

REVIEWER	Sarah Nevitt University of Liverpool, United Kingdom
REVIEW RETURNED	25-Jul-2019

GENERAL COMMENTS	I have conducted a statistical review of the manuscript "A Randomised Controlled Pilot Feasibility Trial of an Early Intervention Programme for Young Infants with Neurodevelopmental Impairment in Uganda: Study Protocol" I found this protocol very interesting and I wish the trial team good luck in this pilot feasibility trial. I understand from the ISRCTN registry entry that this trial is ongoing and therefore any changes to design, recruitment strategy etc. would not be feasible. I do not consider that there are any major issues with this protocol or the trial design. All of my comments are minor and for clarification, or suggestions for the final statistical analysis plan or for conduct of a larger trial. 1) Throughout the manuscript, the authors refer to quality of life and other outcomes being measures at 6 months after recruitment (corresponding to completion of EIP in the intervention arm). As the EIP programme requires 6-10 families to be randomised before it is run, I assume there will be a delay for some randomised participants between randomisation and receiving the intervention (and therefore completing the intervention). So the outcomes will not be measured at the same time point after recruitment or randomisation for all participants? For this reason, I suggest it may be more accurate and more informative to word the measurement times in terms of the age first and time after recruitment in brackets afterwards: e.g. "Participants in both arms will be assessed by study staff masked to trial allocation at two time points; at age 12-17 months (which corresponds to completion of the EIP in the intervention arm, approximately 6 months after recruitment) and again at age 18-23 months (approximately 12 months after recruitment)." 2) Page 5 footnote - I found the use of this footnote is quite confusing - I first read this as reference 1 (i.e. the WHO reference
---

	below). I suggest asking for editorial advice from BMJ Open for how to reference a submitted paper or just state that cost-effectiveness will be examined in a separate protocol. This comment also applied to the footnote on page 12. 3) Primary objective 1: Does this also include acceptability of randomisation? i.e. that a participant may not be randomised to the intervention group? Perhaps edit the first hypothesis to “It will be feasible to conduct and RCT of EIP versus SC in rural and urban contexts and acceptable to families and the community.”? 4) Primary objective 2: I don’t think the abbreviation SC has been previously defined? 5) Page 6, Participants and Recruitment: The sentence about the SPIRIT diagram implies that participants have already been through all stages of this trial. Perhaps reword to make it clear that Figure 1 is the template or proposed structure of what the eventual diagram will look like. 6) Page 6, line 49-50: “Caregivers of infants screening positive on MDAT will be invited to an appointment for written informed consent, and comprehensive neurodevelopmental assessment by study staff ...” I presume that the neurodevelopmental assessment only occurred if written informed consent was given? 7) Page 7, baseline characteristics:” Infant and caregiver demographic information will be recorded at baseline.” Please provide a brief list of the items which will be collected 8) Page 7, Randomisation: Please provide the version of Stata software which will be used 9) Page 7, Randomisation: MRC/ UVRI – I assume this is the institution of the trial management team? I suggest writing out the institutions used within the text once in full before using abbreviations 10) Page 10, Standard Care Arm: I am wondering how close the communities of families recruited into this study may be; for example is it likely that families recruited to the intervention and control groups may know each other and whether there is any risk of the families within the intervention group passing on the knowledge gained to families they know within the control group? If this could happen consider if such a ‘contamination’ effect could be minimised or measured? 11) Page 10, Outcomes: Figure 2 is referred to in this paragraph and I think it should be Figure 1? 12) Page 10, Outcomes: “Inter-rater reliability will be assessed” So does this mean that more than one person will perform each assessment? Inter-rater reliability can only be measured where two or more individuals are collecting the same information
--	--

	13) Page 12: data management and access: "...in-depth interviews (IDIs) and focus groups (FGDs) ..." These abbreviations are defined above in qualitative methods so do not need to be defined again here 14) Page 13, Sample Size: "... assuming a mean PedQL score of 65 in the standard care arm and SD of 20 in both arms." Please provide a reference or source (e.g. clinical expert opinion) for these figures 15) Page 13, Statistical Analysis: "Recruitment and randomisation feasibility will be demonstrated if the target sample size of 126 is achieved." It may also be helpful to examine the number randomised out of the number eligible invited for written consent as a measure of willingness for randomisation or take part in research. 16) Page 13, Statistical Analysis: "Protocol violation rate will be calculated as the number of participants for whom a protocol violation occurs..." Could more than one protocol violation occur? Should this be worded as the number of participants for whom one or more protocol violations occur? 17) Page 13, Statistical Analysis: "For participants in the EIP trial arm, the overall number of modules attended will be summarised using median, range and interquartile range. " Given that there are 10 modules, it may be more informative to summarise the proportions who attend a specific number of session e.g. how many attend all 10? How many attend 9? Or less etc.?" 18) Page 13, Statistical Analysis: "Analysis will be on an intention-to-treat basis and missing data will not be imputed." Missing data could be a problem for some of these measures as it could be informative. E.g. if a total score is calculated from a number of questions on a questionnaire, a missing value treated as a zero could be informative. Consider the measures being used e.g. the PedsQL and whether there are any methods built into the manuals for these measures for accounting for or imputing missing data 19) Page 14, Trial management, data monitoring and reporting of adverse events: "The DSMB will have access to all data on request." Does this include unblinded data? 20) Page 15, Ethics: Paragraph starting "All recruited children will receive SC at the study sites..." I am not sure that this content should be under ethics? Maybe it should be under the Standard Care arm section? 21) Page 16, Online supplementary information: This documents aren't provided as part of this protocol? 22) Figure 1: 12-17 months and 18-23 months should say 'of age'
--	---

VERSION 1 – AUTHOR RESPONSE

Reviewer: 1

Reviewer Name: Maiken Pontoppidan

Institution and Country: The Danish Center for Social Science Research, Denmark

Please state any competing interests or state 'None declared': None declared

Please leave your comments for the authors below

This is a trial protocol for a study examining feasibility, acceptability and the effects of the early intervention programme ABAaNa for parents of infants with a neurodisability in Uganda. The protocol is well written but tends to be a bit long with some repetitions. The protocol does not appear to be rewritten to a journal paper. E.g. there is no abstract and there is a content list”.

Response: We would like to thank the reviewer for their comments however we believe that they have inadvertently reviewed the full study protocol rather than the protocol paper. The submitted paper did indeed include an abstract. The content list that the reviewer refers to is in the trial protocol document and not the submitted manuscript.

“I do not find any reference to a registration in a trial register. Has the trial been registered prior to recruitment?”

Response: Again, I think some confusion has occurred between the trial protocol and the submitted manuscript. The trial registration is clearly stated at the end of the abstract (manuscript, page 2)

“SPIRIT checklist: please make sure that the checklist refers to the right pages. E.g. it says that trial identifier and registry name is on page 2, Sample size calculation is on p 25 (12-13 in the checklist); analysis plan is on p 26 (13 in the checklist)”

Response: The SPIRIT checklist is in fact correct. The page numbers refer to the submitted manuscript not to the trial protocol.

“There are some repetitions - e.g. section 5.1.3 is a longer version of 3.4; section 5.2.4 is a longer version of an earlier; section 5.3 is a longer veriosn of 3.6.3 - It would be easier if it is merged so all the information is at one place”.

Response: I believe this refers to the trial protocol not the submitted manuscript

“Analysis: do you plan to do any imputation of missing data?”

Response: ‘Missing data will not be imputed’ was stated in the submitted manuscript on page 13.

Minor issues:

Please add the references to paragraphs 2, 3 and 4 on p 7, 4 on p 9, 3 on p 10,
P 10 paragraph 3: Are there no never or ongoing studies? Maybe some of the studies by Lynne Murray and Mark Tomlinson could be relevant?

All tables: please spell out abbreviations in a note. E.g. Table 1 DQ and HINE.

All tables: please indicate if high or low score is good

Table 2: what is N? can you add effect sizes?

p 19 3.6.1 QOL - is it parent report?

P 19 3.6.2 How will outcomes be assessed?

Response: I believe these comments refer to the trial protocol not the submitted manuscript Analysis will be on an intention-to-treat basis and missing data will not be imputed. All references are included. Abbreviations are spelt out underneath the figure, table 2 is not included in the paper. QOL is assessed by structured interview conducted by study staff and this is clearly stated in the manuscript (Page 11).

“P 22: 5.2.6, 5.2.7, 5.2.8 who does the assessment? interview or self-report?”

Response: I believe these comments refer to the trial protocol not the submitted manuscript however we agree that this could be better clarified in the paper and have done so by the addition of the statement ‘by interview’.

“There is no actual status of the trial. According to the timeline recruitment ended in 2018 - if this is true the trial is not currently recruiting”.

Response: Again, I think that this comment relates to the trial protocol. Fieldwork for the trial is ongoing in accordance with requirement for submission of a protocol manuscript to BMJ Open. Trial status was not included in the manuscript as will be dependent on the date of publication of the protocol paper.

“Please add line numbers”

Response: The addition of line numbers was not a requirement under the guidance for authors and so was not included. It is not standard practice to include line numbers in the trial protocol which I believe was the document mistakenly reviewed.

“Please add information on how training of the persons doing the intervention will be carried out.”

Response: Training of persons doing the intervention is described in the manuscript on pages 8 & 9 as follows “Facilitators of the group sessions are ‘expert parents’, themselves parents of children with NDI, who have undergone five days of core training followed by regular supervision, face-to-face mentoring meetings and telephone discussions with existing in-country Master Facilitators (trained therapists in Uganda)”

P 15 paragraph 5: who does the screening?

Response: The screening assessments are performed by trained study staff. This has now been clarified on page 6 of the manuscript under the section ‘screening for eligibility’.

Please add expected effect size to sample size calculation p 25

Response: This information was included in the sample size section (page 13).

P 26 qualitative data - please add reference to thematic framework approach and add a bit more information on which areas or themes you expect to analyse

Response: Information on qualitative analysis is stated on page 12, we have augmented this with an additional sentence at the end of the second paragraph below, so that our approach to analysis is clearer:

” Amongst intervention arm families, qualitative techniques will be used to capture information on the feasibility, acceptability and impact of the EIP intervention using qualitative tools including focus group discussions (FGDs), in-depth-interviews (IDIs) and observation.

We will describe the experiences of children and caregivers relating to the intervention received including the impact of the disability, parental confidence level, inclusion in community life and experience of stigma and discrimination. We will examine changes in these domains over the follow-up period and explore attributions of change. In addition, we will perform social mapping of parent networks and group discussions with staff on their perspectives and experiences of using the EIP. The themes guiding our analysis will be drawn not only from objectives of the trial but also from the data, should additional areas of interest emerge during interviews and discussions.”

Reviewer: 2

Reviewer Name: Sarah Nevitt

Institution and Country: University of Liverpool, United Kingdom

Please state any competing interests or state 'None declared': I am working on a trial which has some similarities in the design and setting to this trial (ISRCTN20474555).

I do not consider this to be a conflict of interest, but I am declaring this for transparency

I have conducted a statistical review of the manuscript "A Randomised Controlled Pilot Feasibility Trial of an Early Intervention Programme for Young Infants with Neurodevelopmental Impairment in Uganda: Study Protocol"

I found this protocol very interesting and I wish the trial team good luck in this pilot feasibility trial.

I understand from the ISRCTN registry entry that this trial is ongoing and therefore any changes to design, recruitment strategy etc. would not be feasible. I do not consider that there are any major issues with this protocol or the trial design. All of my comments are minor and for clarification, or suggestions for the final statistical analysis plan or for conduct of a larger trial.

Response: We would like to thank the reviewer for these supportive comments and thoughtful consideration of this feasibility trial protocol and any future larger trial.

1) Throughout the manuscript, the authors refer to quality of life and other outcomes being measures at 6 months after recruitment (corresponding to completion of EIP in the intervention arm). As the EIP programme requires 6-10 families to be randomised before it is run, I assume there will be a delay for some randomised participants between randomisation and receiving the intervention (and therefore completing the intervention). So the outcomes will not be measured at the same time point after recruitment or randomisation for all participants?

For this reason, I suggest it may be more accurate and more informative to word the measurement times in terms of the age first and time after recruitment in brackets afterwards: e.g.

"Participants in both arms will be assessed by study staff masked to trial allocation at two time points; at age 12-17 months (which corresponds to completion of the EIP in the intervention arm, approximately 6 months after recruitment) and again at age 18-23 months (approximately 12 months after recruitment)."

Response: The outcomes are measured at two time points, 6 and 12 months after recruitment as identified by the reviewer. This has now been clarified on page 10 as per the reviewer's recommendation.

2) Page 5 footnote - I found the use of this footnote is quite confusing - I first read this as reference 1 (i.e. the WHO reference below). I suggest asking for editorial advice from BMJ Open for how to reference a submitted paper or just state that cost-effectiveness will be examined in a separate protocol. This comment also applied to the footnote on page 12.

Response: We agree that this appears confusing and have therefore deleted the footnote from the revised manuscript. Reference to the separate protocol paper for the economic evaluation has been retained on pages 5 and 12 and we would be grateful for the advice of the editorial team as to how they would like us to reference this in any final publication.

3) Primary objective 1: Does this also include acceptability of randomisation? i.e. that a participant may not be randomised to the intervention group? Perhaps edit the first hypothesis to "It will be feasible to conduct and RCT of EIP versus SC in rural and urban contexts and acceptable to families and the community."?

Response: We agree with the reviewer that the primary outcome of interest is stated as "Feasibility of participant recruitment and randomisation as assessed by the total number recruited and randomised

to each arm” and therefore does include feasibility of randomisation. The manuscript has therefore been amended according to the reviewer’s recommendation (page 5, Hypothesis 1).

4) Primary objective 2: I don’t think the abbreviation SC has been previously defined?

Response: The abbreviation SC has now been defined in objective 1.

5) Page 6, Participants and Recruitment: The sentence about the SPIRIT diagram implies that participants have already been through all stages of this trial. Perhaps reword to make it clear that Figure 1 is the template or proposed structure of what the eventual diagram will look like.

Response: Thank you for this recommendation, the word ‘planned’ has now been added to improve clarity as per the reviewer’s advice.

6) Page 6, line 49-50: “Caregivers of infants screening positive on MDAT will be invited to an appointment for written informed consent, and comprehensive neurodevelopmental assessment by study staff ...” I presume that the neurodevelopmental assessment only occurred if written informed consent was given?

Response: Yes, neurodevelopmental assessment would only occur if written parental consent has been provided. This has now been clarified in the text by the addition of ‘, and if provided, ..’

7) Page 7, baseline characteristics:” Infant and caregiver demographic information will be recorded at baseline.” Please provide a brief list of the items which will be collected

Response: Clarification of baseline characteristics has now been clarified by the inclusion of “including date of birth, age, sex, birth order, parity, antepartum, intrapartum and postpartum history, family and medical history, developmental history, mother’s education and occupation, family details including family size, and ages, household incomes, household SES and residence”

8) Page 7, Randomisation: Please provide the version of Stata software which will be used

Response: The version of Stata used has now been clarified in the text (version 15)

9) Page 7, Randomisation: MRC/ UVRI – I assume this is the institution of the trial management team? I suggest writing out the institutions used within the text once in full before using abbreviations

Response: The full formal name of the organisation (MRC/UVRI & LSHTM Uganda Research Unit) has now been added for clarity

10) Page 10, Standard Care Arm: I am wondering how close the communities of families recruited into this study may be; for example is it likely that families recruited to the intervention and control groups may know each other and whether there is any risk of the families within the intervention group passing on the knowledge gained to families they know within the control group? If this could happen consider if such a ‘contamination’ effect could be minimised or measured?

Response: Thank you for this observation – this was an unintended omission from the manuscript. Contamination of the control arm will be monitored and reported, and this has now been clarified in the text on page 10.

11) Page 10, Outcomes: Figure 2 is referred to in this paragraph and I think it should be Figure 1?

Response: Apologies for this oversight, the reviewer is correct, and this has now been amended in the text.

12) Page 10, Outcomes: “Inter-rater reliability will be assessed” So does this mean that more than one person will perform each assessment? Inter-rater reliability can only be measured where two or more individuals are collecting the same information

Response: The process for assessing inter-rater reliability has now been clarified in the text as follows (page 10), “Two assessors will independently assess a small proportion of the children and inter-rater reliability will be calculated.”

13) Page 12: data management and access: "...in-depth interviews (IDIs) and focus groups (FGDs) ..." These abbreviations are defined above in qualitative methods so do not need to be defined again here

Response: Thank you for this observation. The abbreviations alone are now used in this section.

14) Page 13, Sample Size: "... assuming a mean PedQL score of 65 in the standard care arm and SD of 20 in both arms." Please provide a reference or source (e.g. clinical expert opinion) for these figures

Response: These figures are based on findings from pilot work. This has now been clarified in the text with the addition of the following statement (page 13), "Assumptions are based on data from the pilot study showing a mean caregiver PedQL score for families before the intervention of 64.9 (standard deviation (SD) 19.6) and mean score of 78.9 for families after receiving the intervention (SD 17.5)."

15) Page 13, Statistical Analysis: "Recruitment and randomisation feasibility will be demonstrated if the target sample size of 126 is achieved." It may also be helpful to examine the number randomised out of the number eligible invited for written consent as a measure of willingness for randomisation or take part in research.

Response: Thank you for this suggestion. Data on the number randomised out of the number eligible invited for written consent will be reported as part of the CONSORT flowchart.

16) Page 13, Statistical Analysis: "Protocol violation rate will be calculated as the number of participants for whom a protocol violation occurs..." Could more than one protocol violation occur? Should this be worded as the number of participants for whom one or more protocol violations occur?

Response: Thank you for this observation. The text has been amended according to the reviewer's suggestion (page 13).

17) Page 13, Statistical Analysis: "For participants in the EIP trial arm, the overall number of modules attended will be summarised using median, range and interquartile range. " Given that there are 10 modules, it may be more informative to summarise the proportions who attend a specific number of session e.g. how many attend all 10? How many attend 9? Or less etc.?

Response: We agree that this would be a sensible approach and have adopted the reviewer's suggestion (page 13).

18) Page 13, Statistical Analysis: "Analysis will be on an intention-to-treat basis and missing data will not be imputed." Missing data could be a problem for some of these measures as it could be informative. E.g. if a total score is calculated from a number of questions on a questionnaire, a missing value treated as a zero could be informative. Consider the measures being used e.g. the PedsQL and whether there are any methods built into the manuals for these measures for accounting for or imputing missing data.

Response: Each of the tools used have mechanisms for managing missing data in the standardised protocols for their use. For instance, for PedsQL the denominator used to calculate the score reflects the number of questions actually answered, not the number of questions presented. In our pilot study, we had very few participants with any missing data for individual elements of each measure, so do not anticipate that this will impact.

19) Page 14, Trial management, data monitoring and reporting of adverse events: "The DSMB will have access to all data on request." Does this include unblinded data?

Response: Yes, access to unblinded data will be shared with the DSMB if requested.

20) Page 15, Ethics: Paragraph starting "All recruited children will receive SC at the study sites..." I am not sure that this content should be under ethics? Maybe it should be under the Standard Care arm section?

Response: This has been included under the ethics section for clarity that standard care was not withheld to intervention arm participants as this would not be considered ethical.

21) Page 16, Online supplementary information: These documents aren't provided as part of this protocol?

Response: These documents can be made available as online supplementary information and have been uploaded with the manuscript as part of this resubmission.

22) Figure 1: 12-17 months and 18-23 months should say 'of age'

Response: This has been added to Figure 1.

VERSION 2 – REVIEW

REVIEWER	Maiken Pontoppidan VIVE - the Danish Center for Social Science Research Denmark
REVIEW RETURNED	27-Aug-2019

GENERAL COMMENTS	I am very sorry that I reviewed the original protocol and not the paper version in the first round. I must have mixed them up by accident. The paper is very well written and I only have a few minor details: P 4: Can you add the references to the second paragraph on the impairments? P 7: Inclusion and exclusion criteria: What if a child is exactly 12 months old - are they included or not? inclusion is 6-11 months, exclusion is >12 months P 16: Insert a line break before Competing interest statement Good luck with the trial!
---

REVIEWER	Sarah Nevitt University of Liverpool United Kingdom
REVIEW RETURNED	16-Aug-2019

GENERAL COMMENTS	Thank you to the reviews for their responses and for their efforts addressing my statistical comments. I am satisfied that all of my comments have been addressed and happy to recommend this protocol for publication.
--

VERSION 2 – AUTHOR RESPONSE

Reviewer: 2

Thank you to the reviews for their responses and for their efforts addressing my statistical comments. I am satisfied that all of my comments have been addressed and happy to recommend this protocol for publication.

Response: Many thanks to the reviewer for her comments and recommendations.

Reviewer: 1

I am very sorry that I reviewed the original protocol and not the paper version in the first round. I must have mixed them up by accident.

The paper is very well written and I only have a few minor details:

P 4: Can you add the references to the second paragraph on the impairments?

Response: This sentence has now been referenced

P 7: Inclusion and exclusion criteria: What if a child is exactly 12 months old - are they included or not? inclusion is 6-11 months, exclusion is >12 months

Response: Thank you for this helpful comment. This has now been clarified in the text that inclusion is 6-11 completed months and children 12 months of age or older will be excluded.

P 16: Insert a line break before Competing interest statement

Response: This was an oversight, thanks for pointing it out. This has been amended.

Good luck with the trial!

Thank you!